# Water-generated dangling linkers in a metal-organic framework

Yao Fu[1,2], Yifeng Yao[3], Subhradip Paul [1], Kenji Mochizuki [3] ✉ &
Gaël De Paëpe [1] ✉

Metal-Organic Frameworks (MOFs) have attracted widespread attention for their applications in water-related contexts. A comprehensive understanding of the molecular-level interactions between water and MOFs is crucial for guiding molecular design and optimizing water-related applications. Water can act as a passive guest, interacting weakly with open metal sites or polar linkers without altering the framework, or as a reactive species that cleaves the dative bonds between inorganic clusters and organic linkers, leading to irreversible degradation. In this work, we uncover a significant impact of water on the metal-linker linkage in UiO-66, a prototype MOFs which is considered highly stable with water. The adsorption of water molecules in UiO-66 results in the displacement of firmly attached carboxylate groups of the linker, thereby transforming them into dangling carboxylate groups. These dangling groups are stabilized by water molecules and $\mu_3$-OH through hydrogen bonding. Remarkably, this structural transformation is reversible upon water removal. These findings were elucidated through the integration of multi-dimensional solid-state NMR, cutting-edge dynamic nuclear polarization (DNP) techniques, and computational calculations. By challenging conventional wisdom, our research has introduced a reversible molecular structure evolution scenario, redefining the understanding of water-MOF interactions.

Metal-organic frameworks (MOFs) have recently emerged as highly effective water sorbents[1,2], unlocking a multitude of applications such as atmospheric water harvesting[3–5], adsorptive heat transformation[6–8], autonomous indoor humidity control[9–11]. Additionally, MOFs serve versatile roles as drug delivery carriers[12], energy devices[13], and sensors[14] within aqueous environments. The success of above applications prompts a thorough investigation into the molecular-level interactions between water and MOFs. Researchers have employed various techniques, including solid-state NMR[15–21], which has proven especially useful in studying MOFs, as well as diffraction[22,23], Raman[24], infra-red[25,26] to explore the water structure within MOFs. Notably, Yaghi group and co-authors have made significant progress by employing single X-ray diffraction to decipher the detailed water-filling mechanism, revealing step-by-step water-binding structures[27].

Water can behave very passively, interacting weakly with open metal sites[23,26] or polar organic linkers[27] within MOFs without causing significant structural changes. However, under certain conditions, it can also become highly reactive, cleaving the dative bonds between the inorganic building units and the organic linkers and often leading to irreversible framework degradation[28].

In this study, we uncovered the active role of water, demonstrating its ability to induce dangling linkers within a zirconium-based MOF, UiO-66 ($Zr_6O_4(OH)_4(BCD)_6$, BDC = 1,4-benzenedicarboxylate)[29]. UiO-66 has been widely employed in diverse water-related applications, including water treatment[30,31], water splitting[32], and molecular delivery[12,33]/separation[34] in aqueous solutions. Up to now, UiO-66 has been regarded as a stable material in aqueous environments, supported by water adsorption isotherm[22,35] and powder X-ray diffraction

[1]Univ. Grenoble Alpes, CEA, IRIG-MEM, Grenoble, France. [2]Department of Chemistry, Fudan University, Shanghai, PR China. [3]Department of Chemistry, Zhejiang University, Hangzhou, PR China. ✉e-mail: kenji_mochizuki@zju.edu.cn; gael.depaepe@cea.fr

measurements[34]. However, recent studies by the Pourpoint group have indicated the vulnerability of Zr-O bonds, suggesting that UiO-66 may exhibit lower stability than previously assumed[36]. Our research further demonstrated that water could break the linker-metal bonds in UiO-66 under ambient conditions, leading to the formation of dangling linkers. Moreover, water molecules not only bonded to the metal sites but also played a crucial role in stabilizing these dangling linkers via hydrogen bonding. This discovery of water-induced dangling linkers was revealed through multidimensional solid-state NMR, dynamic nuclear polarization (DNP) techniques, and computational calculations.

## Results and discusion

The ideal UiO-66 sample in this study was synthesized using well-established methods[37]. When the MOF adsorbs a certain amount of water, it becomes crucial to measure the water content within the MOFs. Solid-state $^1H$ NMR is a powerful technique for this purpose and has been utilized to quantify the water content in porous materials[38-40]. Two approaches were employed for water adsorption: either by exposing the MOF sample to ambient air for several days or by artificially adding water into the MOF powders directly. We examined $^1H$ magic angle spinning (MAS) spectra for samples with varying water loading amounts (Fig. 1a). As expected, the spectra feature mainly signals[18] from aromatic protons of the $BDC^{2-}$ linker (~8 ppm), $\mu_3$-OH protons from the Zr-O cluster (0–3 ppm), and signals from adsorbed water. Interestingly, chemical shift corresponding to adsorbed $H_2O$ varies from 4 to 6 ppm depending on the amount of $H_2O$ adsorbed.

To ensure quantitative results, a recycle delay of 2–3 s (slightly longer than 5× the spin-lattice relaxation time, $T_1$, Table S1) was used. Spectral deconvolution (see example from Fig. S1) facilitated the naming of samples as "x $H_2O$ /u.c." where x = the number of $H_2O$ molecules in the MOF unit cell $Zr_{24}O_{16}(OH)_{16}(BDC)_{24}$. The "fresh" sample refers to the synthesized sample activated under 120 °C. The "dried" sample is obtained by heating the "320 $H_2O$ /u.c." sample overnight to almost eliminate all adsorbed water. As evident from the spectra, in both the "fresh" and "dried" samples, the water content in the pores is negligible. We conducted powder X-ray diffraction for all the samples (Fig. S2), and they displayed similar patterns, indicating that the samples retained long-range order. This observation is consistent with past research where UiO-66 was considered highly stable under water environments[34].

Nevertheless, such conclusion is contradicted by the $^{13}C$ NMR spectra acquired through $\{^1H\}$-$^{13}C$ cross-polarization (CP). As seen in Fig. 1c, for the fresh UiO-66 sample, there are three characteristic peaks. The peak at 172 ppm corresponds to the carboxylate group (labeled as C1), while peaks C2 and C3 correspond to the -C and -CH carbons from the benzene ring. When there are few waters in one unit cell (26 $H_2O$/u.c.), the $^{13}C$ spectrum looks similar to the fresh sample. However, with an increase in water content up to 160 per unit cell, all peaks broaden, a new peak (labeled as C4) emerges to the left of C1, and C3 exhibits two components. These observations collectively suggest alterations in ligand dynamics, along with structural changes within the ligand itself. This pattern persists when the water content is doubled (320 $H_2O$/u.c.). Upon the removal of water from the cages, the structural changes revert, indicating a reversible transition between short-range order and disorder in the structure during water adsorption. Notably, this reversible transition remains stable even under prolonged aqueous conditions for up to one year (Fig. S3).

Based on this new finding, revealing the emerging disordered structure, and understanding the role of $H_2O$ in the UiO-66 become quite important. $^1H$-$^{13}C$ two-dimensional (2D) heteronuclear correlation (HETCOR) method was employed on the 160 $H_2O$/u.c. sample, targeting the local environment surrounding the newly identified C4 and its interaction with $H_2O$ molecules (Fig. 2b, Fig. S4). Under a mixing time of 5 ms, the HETCOR spectrum reveals contribution from several

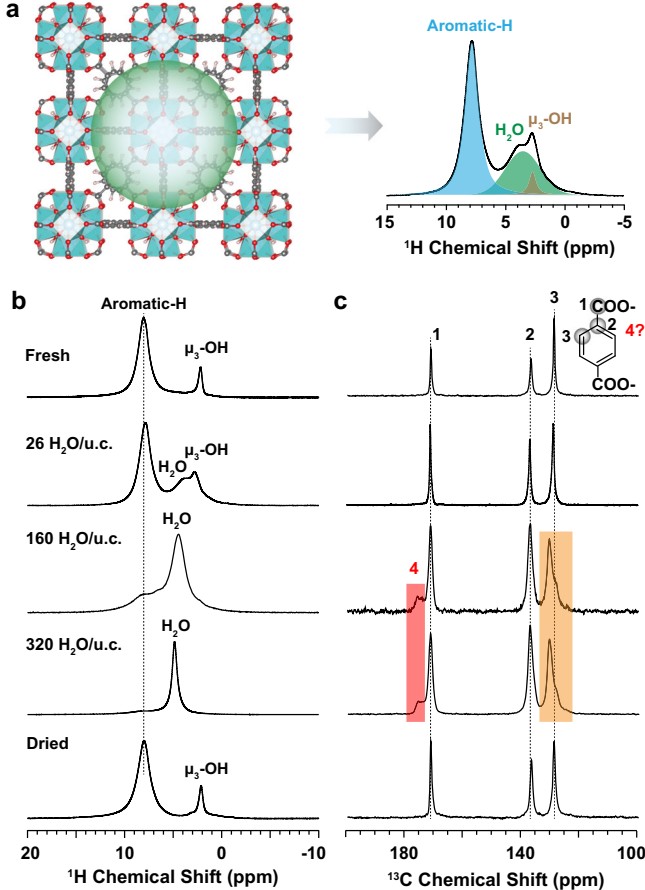

**Fig. 1 | Reversible water-driven structural transitions uncovered by solid-state NMR. a** An illustration demonstrating the use of solid-state NMR to quantify adsorbed water content in MOFs. On the left side, a depiction of water adsorption in an ideal UiO-66 unit cell (u.c.), with water represented by green balls. On the right side, the $^1H$ spectrum of UiO-66 after water adsorption. The quantity of water in a single unit cell of the MOF is assessed by comparing signal intensities between $H_2O$ protons (green peak) and aromatic protons (blue peak). **b** $^1H$ one-pulse spectra and **c** $^{13}C$ cross-polarization (CP) spectra obtained from a series of UiO-66 samples with variable water contents, recorded under spinning rate at 10 or 15 kHz. A newly emerged C4 area and an enlarged C3 area upon water adsorption are highlighted in the red and orange shaded regions, respectively.

$^1H$ peaks, including two distinct $H_2O$ environments from bulk $H_2O$ (~4.7 ppm) and hydrogen-bonded surface water (~6.2 ppm), as well as aromatic $^1H$ from the linkers at ~8.0 ppm. In addition, one can also observe several $^1H$ resonances in the 0–3 ppm range, which are tentatively assigned to the $\mu_3$-OH resonances. These resonances are clearly confirmed on the 1D $^1H$ NMR spectrum of the 160 $H_2O$/u.c. sample recorded at higher MAS frequencies (Fig. S5), which provides improved spectral resolution. It is interesting to note that the 0–3 ppm resonances, assigned to $\mu_3$-OH species, vary with the amount of water content, but that the fresh and dried samples mostly show a single $\mu_3$-OH resonance at ~2 ppm (Fig. 1b). In absence of $H_2O$ molecules in the pore, all $\mu_3$- OH are equivalent (four by metal cluster) and thus resonate at the same frequency (2 ppm). The changes in the $\mu_3$-OH resonances suggest the formation of detached linkers, likely due to the displacement of carboxylate linkers from the metal clusters, which can influence the chemical shift of the remaining $\mu_3$-OH groups.

In addition, the HETCOR spectrum (Fig. 2b) reveals that all linker carbon atoms, including C4, exhibit correlation peaks with bulk $H_2O$, hydrogen-bonded surface $H_2O$ or aromatic protons. Finally, it is interesting to note the absence of cross-peaks between C4 and

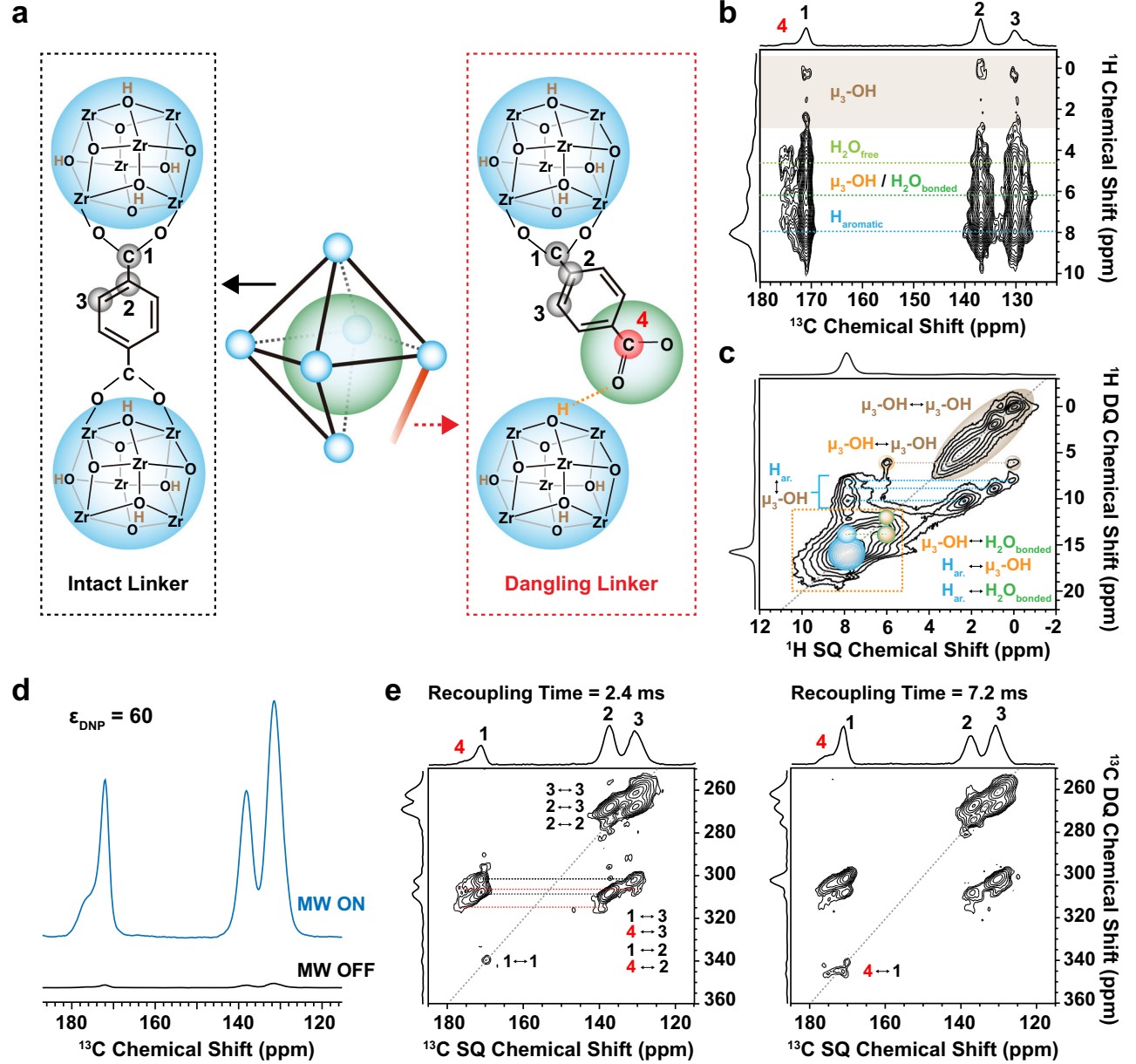

**Fig. 2 | Identification of a water-induced dangling linker by multidimensional NMR. a** Illustration of the intact linker and dangling linker of UiO-66. Blue balls represent the zirconium-oxide cluster, while the green balls represent the water clusters. All the carbons from the linkers are labeled. Most $\mu_3$-OH protons are labeled in brown, whereas the $\mu_3$-OH proton associated with the C4 site is labeled in orange. **b** $^1$H–$^{13}$C HETCOR NMR spectrum measured at room temperature on the "160 $H_2O$/u.c." sample with a mixing time of 5 ms. **c** $^1$H-$^1$H 2D double-quantum single-quantum (DQ-SQ) correlation spectrum acquired at room temperature on the "160 $H_2O$/u.c." sample with a recoupling time of 57 μs. The dotted orange region shows spatial correlations arising from the proximity of aromatic protons, the $\mu_3$-OH group near the detached linker, and hydrogen-bonded water, along with their respective self-correlations. **d** Illustration of the signal enhancement from DNP via 1D {$^1$H-}$^{13}$C CPMAS spectra of UiO-66 sample, recorded with (blue) and without (black) microwave irradiation at 100 K. **e** $^{13}$C-$^{13}$C 2D DQ-SQ correlation spectra of UiO-66 sample recorded at 100 K with recoupling times = 2.4 ms and 7.2 ms.

isolated $\mu_3$-OH resonances (in the 0–3 ppm range) at 5 ms HETCOR spectrum. This indicates that water molecules cluster around linkers and that the C4 site is probably associated with hydrogen-bonded water. Interestingly, the chemical shift of aromatic proton near C4 (7.6 ppm) is slightly lower than that of the aromatic proton on an intact linker (8.0 ppm), a phenomenon explained by DFT calculations below.

To investigate this further, 2D $^1$H-$^1$H double-quantum single-quantum (DQ-SQ) NMR experiment[41] with short-recoupling times (57 μs and 171 μs) were performed on the 160 $H_2O$/u.c. sample (Fig. 2c, S6, and S7). These experiments reveals three distinct $\mu_3$-OH species

(labeled in brown), appearing in the 0–3 ppm range, that are not in close spatial proximity to each other. It also demonstrates spatial correlations between all original $\mu_3$-OH groups and aromatic protons (labeled in blue, ~8 ppm). In addition, a cross-peak is detected between one of the $\mu_3$-OH species and a resonance at approximately 6 ppm. This resonance is tentatively assigned to a $\mu_3$-OH group as well (labeled in orange) associated with the C4 site, as further supported by the DFT calculations described below. We note this $\mu_3$-OH species in orange resonates similarly to hydrogen-bonded water. We propose that the spatial correlations observed in the dotted orange region result from the proximity among aromatic protons, the $\mu_3$-OH group close to the

detached linker, and hydrogen-bonded water, as well as their respective self-correlations. We also note that this interpretation is consistent with the HETCOR data since a spatial correlation between the C4 site and a resonance at ~6 ppm is also present in Fig. 2b, consistent with the presence of a $\mu_3$-OH specie.

All together, these experimental data suggest that the C4 resonance arises from a detached linker coordinating with both water and a $\mu_3$-OH moiety. In order to prove this, and to explore their relative distributions within the MOF structure, $^{13}C$-$^{13}C$ through-space correlation experiments would be highly effective. However, the practicality of such 2D experiments faces hindrance due to the low natural abundance of $^{13}C$ (1.1%), which makes finding an NMR active spin-pair two orders of magnitude lower. To overcome this sensitivity limitation, state-of-the-art DNP techniques[42–49] were utilized. The DNP efficiency ($\varepsilon_{DNP}$) was evaluated by mixing the MOF with a glass-forming matrix d8-glycerol: $D_2O$: $H_2O$ (6:3:1 by volume) containing the polarizing agent AMUPol,[50] resulting in an enhancement factor of 60, when comparing $^{13}C$ CP spectra with and without microwave irradiation (Fig. 2d)[51]. The complete $^{13}C$ spectrum, including signals from the glycerol, is presented in Fig. S8. Since the distance between two nitroxide centers of AMUPol[50] (~12 Å) exceeds the dimensions of the UiO-66 octahedral (~9 Å diameter) and tetrahedral cages (~7 Å diameter), AMUPol cannot penetrate the UiO-66 microcrystals. Nevertheless, hyperpolarization is effectively transferred from AMUPol to the protons within the MOFs via $^1H$-$^1H$ spin diffusion.

The significant enhancement renders natural-abundance $^{13}C$-$^{13}C$ 2D DQ-SQ experiments feasible, allowing for the acquisition of a good signal-to-noise spectrum within 15 hours. Notably, under a short recoupling time of 2.4 ms using the S3 dipolar recoupling sequence[52,53], correlation peaks indicate short-range C-C proximity within 2–3 Å (Fig. 2d, S9a). The short-range correlation primarily reveals intramolecular peaks for the BDC$^{4-}$ ligand, denoted as C1-C2, C1-C3, C2-C3, etc. (Fig. S9b). Intriguingly, strong correlations C4-C2, C4-C3 also emerge, indicating that the C4 peak originates from the linker carboxylate carbon. This implies the existence of two distinct environments for the BDC$^{4-}$ linker carboxylate group: one where the group remains firmly bounded to metal-oxide sites, referred as the intact linker; and another where it may detach from these sites, due to the presence of water clusters, referred as the dangling linker (as illustrated in Fig. 2a). The concept of dangling linker was proposed during the post-synthetic linker exchange in UiO-66, primarily based on computational insights[54], and has not yet been confirmed by experimental data. With a longer recoupling time of 7.2 ms, C1-C4 correlations are also observed. Extending the recoupling time further, up to 30 ms (Fig. S10), reveals C-C proximities up to 8 Å (as simulated in Fig. S9a). Importantly, no self-correlated peaks from dangling carboxylate groups emerge, indicating that the dangling linkers in UiO-66 are primarily disconnected on only one side rather than both sides. This also suggests these dangling linkers are likely homogeneously distributed throughout the framework, rather than clustered in localized regions.

After meticulously gathering structural insights from the previous experiments, it becomes apparent that dangling carboxylate groups within the MOF are enveloped by water molecules. This rearrangement results in the generation of higher chemical shifts compared to firmly attached carboxylate group. To further explore our experimental findings and discern the possible configurations of these dangling carboxylate groups, we conducted quantum mechanical calculations to compare the energy of formation ($\Delta E$) for various configurations originating from the attached carboxylate group. The reaction we considered is:

$$UiO66 + n \cdot H_2O \rightarrow UiO66 \cdot nH_2O (n = 1 \text{ or } 2) \Delta E$$

Where n represents the number of water molecules directly interacting with the carboxylate groups, resulting in various dangling linker configurations with the remaining post-reaction small molecules (i.e. $H_2O$ or $OH^-$) occupying the uncoordinated metal sites. Two cases were considered here: (A) the dangling linker points toward the closest $\mu_3$-OH, and (B) the dangling linker points away from the closest $\mu_3$-OH. Based on charge balance, several dangling linker configurations can be conceived for both cases (Figs. 3a, b, S11, S12): (I) $COO^- + 2H_2O$; (II) $COOH + OH^- + void$; (III) $COO^- + H_2O$; (IV) $COOH + OH^- + H_2O$; and (V) $COOH + OH^-$. Here, configurations with a positive $\Delta E$ are considered unlikely to exist, while those with a more negative $\Delta E$ are presumed to be more stable. Our investigations revealed that only the (I) $COO^- + 2H_2O$, and (IV) $COOH + OH^- + H_2O$ configurations are stable and could potentially exist in both case (A) and case (B) (Figs. 3a, 3b, S12, S12). Among these, the (I) $COO^- + 2H_2O$ configuration, in which the dangling linker points toward the $\mu_3$-OH group (Case A), exhibits the greatest stability with an energy of $\Delta E = -62.7$ kJ/mol.

To further examine these configurations, we performed $^{13}C$ chemical shift calculations, confirming that for both case (A) and case (B), only the (I) $COO^- + 2H_2O$ combination aligns with our experimental data. This configuration exhibits a ~5 ppm higher chemical shift for dangling carboxylate group (peak h) compared to the intact carboxylate group (peak a) (Fig. 3c). Conversely, for the (IV) $COOH + OH^- + H_2O$ and all other combinations (II, III, V), the dangling carboxylate group's chemical shift is lower or comparable to that of the intact group (Fig. S11, S12). Besides, the absence of a highly deshielded COOH $^1H$ peak (~9–12 ppm) (Fig. 1b, S5), along with the absence of signals corresponding to the C4 and COOH protons in $^1H$-$^{13}C$ 2D experiments (Fig. 2b), enabled us to rule out the possibility of the (IV) $COOH + OH^- + H_2O$ configuration. It is worth noting that the concentration of dangling carboxylate groups within the entire MOF is notably high (30%, as depicted in Fig. S13), making it improbable for the dissolved products of acidic COOH and basic $OH^-$ to exist at such high concentrations.

Lastly, the calculated $^1H$ chemical shifts for the labeled $\mu_3$-OH groups (highlighted in orange in Fig. 3a) in configurations (A-I) and (B-I) were compared. In configuration (A-I), the $\mu_3$-OH group forms a hydrogen bond with the dangling linker, which strengthens its deshielding and shifts its resonance to a higher frequency of 5.9 ppm. This observation supports the absence of a HETCOR correlation peak between the C4 carbon and protons in the 0–3 ppm range in Fig. 2b, and aligns with the appearance of a $\mu_3$-OH resonance at 6 ppm in the $^1H$-$^1H$ 2D NMR spectrum (Fig. 2c). In contrast, configuration (B-I) shows a typical $\mu_3$-OH chemical shift of 1.8 ppm. In this case, the distance between the dangling carboxylate carbon (C4) and the nearest $\mu_3$-OH proton is approximately 4.6 Å, a HETCOR correlation would still be expected, since similar distances between $\mu_3$-OH protons and closet intact linker C3 carbons produce observable correlations (Fig. S14, Fig. 2b). The correlation between C4 and $\mu_3$-OH at 1.8 ppm is however absent in the HETCOR. These findings strongly support (A-I) $COO^- + 2H_2O$ configuration, in which the dangling linker pointing toward the $\mu_3$-OH group, as the most plausible structure. The calculated average aromatic $^1H$ chemical shift of the dangling linker (8.40 ppm) is slightly lower than that of the aromatic $^1H$ in the intact linker (8.47 ppm) (Fig. S15). This difference explains the slightly reduced $^{13}C$-$^1H$ correlation on proton dimension of the dangling linker in the HETCOR spectrum (Fig. 2b).

Upon closer examination of the solved structure A-I in Fig. 3a, The two water molecules occupy the uncoordinated Zr sites and co-stabilize the dangling $COO^-$ group together with the neighboring $\mu_3$-OH group through multiple hydrogen bonds, consistent with previous proposition by molecular dynamics calculations[55]. The water molecules function as intermediates in the metal–linker coordination, exhibiting behavior analogous to the hemilability concept previously described for STAM-17-OEt[28]. Such interactions not only stabilize the dangling carboxylate group but also contribute to the overall structural integrity and properties of the MOF. This finding is further

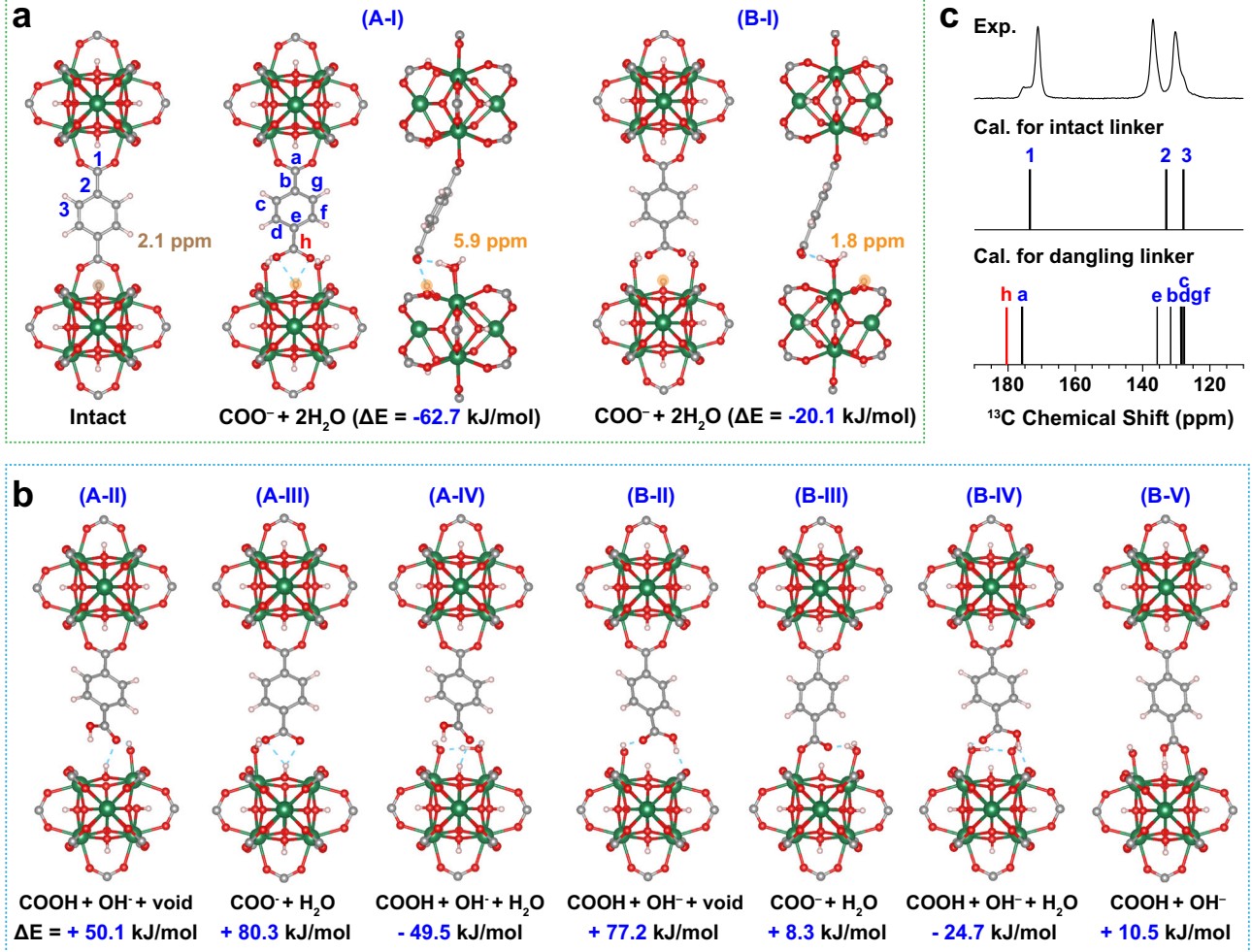

**Fig. 3 | Dangling linker configuration validated by DFT calculations. a, b** The intact linker structure and proposed dangling linker configurations based on case A (where the dangling linker points toward $\mu_3$-OH) and case B (where the dangling linker points away from $\mu_3$-OH) are presented, with all structures optimized through DFT calculations. The energies derived from quantum mechanical calculations are indicated for each configuration. The calculated $^1$H chemical shifts for the labeled $\mu_3$-OH groups (highlighted in brown or orange) in configurations of Intact, (A-I) and (B-I) are shown. Green, red, gray, and white spheres represent Zr, O, C, and H atoms, respectively, while hydrogen bonding is illustrated using blue dashed lines. **c** Experimental $^{13}$C CPMAS spectrum of the "320 $H_2O$/u.c." sample and the calculated $^{13}$C chemical shifts of intact linker and dangling linker structure based on configurations shown in (A-I). All the carbons are labeled in the structure.

supported by FT-IR experiments, which reveal a more disordered structure and changes in the out-of-plane stretching vibration[56] of the carboxylate group upon water adsorption in UiO-66 (Fig. S16).

Our investigation has revealed the significant impact of water molecules on the structural evolution of UiO-66 MOFs. Through meticulous experimentation and analysis, we have observed water induces structural changes, leading to the dissociation of carboxylate groups and the formation of water-stabilized dangling linkers within the framework. These findings provide valuable insights into the complex interplay between water molecules and MOF structures, shedding light on the mechanisms governing their behavior and stability. The reversible structural changes suggest that these MOFs can be effectively used for water adsorption and desorption while maintaining structural integrity. This deeper understanding opens up exciting possibilities for the design and optimization of MOFs tailored for a wide range of water-related applications.

## Methods

### Synthesis of ideal UiO-66

The ideal UiO-66 was synthesized following the Lillerud's recipe[37]. We sequentially added 3.781 g $ZrCl_4$ (16.22 mmol), 2.865 ml 35 % HCl (32.45 mmol), and 5.391 g $H_2BDC$ (32.45 mmol) to a 150 ml Teflon liner containing 97.40 ml N,N'-dimethyl formamide (1258 mmol). After all the reagents dissolved, the liner was sealed in a stainless-steel autoclave and then placed in an oven at 220 °C for 24 hrs. The resulting ideal UiO-66 was washed three times with DMF, three times with methanol for a day each, and three times with deionized water for a day each. The washed products were separated from the solvent by centrifugation, dried under vacuum at 120 °C for 1 day, and grounded with a mortar and pestle. This sample is labeled as "fresh UiO-66". The successful preparation of ideal UiO-66 was confirmed by powder X-ray diffraction and thermogravimetric analysis.

### Water adsorption / desorption in UiO-66

Two approaches were used for water adsorption in fresh UiO-66: (1) exposing the UiO-66 sample to ambient air for several days, or (2) directly adding a known volume of water (e.g., 10 μL to 120 μL) to 50 mg of MOF powder using a pipette. To ensure uniform water adsorption, the mixture was gently ground. Despite the water adsorption, the UiO-66 samples retained a dry appearance. For water desorption, these samples were heated to 120 °C under vacuum

overnight to eliminate almost all the adsorbed water, resulting in the "dried" UiO-66 sample.

## Characterization methods

PXRD were carried out on samples placed on a quartz holder using a Rigaku Ultimate-IV X-ray diffractometer operated at 40 kV/30 mA with Cu Kα line (λ = 1.5418 Å). Patterns were collected in reflectance Bragg-Brentano geometry in the 2θ range from 3 to 50°.

Fourier Transform Infrared (FT-IR) spectra were recorded on a Perkin-Elmer 100 FT-IR spectrometer, fitted with a liquid-nitrogen cooled mercury-cadmium-telluride (MCT) detector. Both the dried and wet UiO-66 samples were analyzed directly, without dilution or mixing with KBr.

## Solid-state NMR and DNP-enhanced NMR

Room temperature $^{13}$C and $^1$H solid-state NMR experiments of fresh / dried and water adsorbed UiO-66 samples were performed on a Bruker Avance III HD 400 MHz NMR spectrometer ($^1$H, 400.13 MHz; $^{13}$C, 100.61 MHz) using a 3.2 mm magic angle spinning (MAS) probe. $^1$H spectra were acquired under the spinning rate of 10-20 kHz, using a one-pulse sequence, with a recycle delay of 2–3 s, which is slightly 5 times longer than their spin-lattice relaxation times. $^{13}$C spectra were collected using either cross-polarization (CP) or direct polarization (DP) sequences under MAS of 15 kHz. The recycle delay for DP sequence was set to 200 s to ensure all the $^{13}$C signals were recovered to equilibrium, while the recycle delay of CP was set to 1.5 s. The $^1$H radio frequency (RF) field strength was 100 kHz and the $^{13}$C RF field strength was 83 kHz. Two-dimensional $^1$H-$^1$H double quantum-single quantum (DQ-SQ) NMR experiments were conducted using a 1.3 mm MAS probe. The experiments employed a rotor-synchronized BABA (Back-to-Back) pulse sequence[41], with the mixing time set to one rotor period. The magic angle spinning (MAS) rate was maintained at 35 kHz. The $^1$H and $^{13}$C signals were referenced to those of adamantane at 1.8 ppm ($^1$H) and 38.5 ppm ($^{13}$C methylene).

For DNP sample preparation, a mixture containing 30 mg of the fresh MOF sample and 45 μL of 10 mM AMUPol solution (comprising 60% d$_6$-glycerol, 30% D$_2$O, and 10% H$_2$O by volume) was transferred into a 3.2 mm zirconia thin rotor. Proton polarization enhancement originating from electron polarization was subsequently transferred to $^{13}$C nuclei via a standard CP step. The efficiency of the DNP process was obtained by comparing $^{13}$C CP spectra with and without microwave irradiation.

For DNP-enhanced $^{13}$C-$^{13}$C double quantum-single quantum (DQ-SQ) experiments were performed on a Bruker Avance III 400 MHz system equipped a low temperature ( ~ 100 K) double resonance 3.2 mm MAS probe. The 2D spectra were recorded at 100 K and a MAS rate of 13.2 kHz. Dipolar recoupling sequence S3[52,53] was used for DQ excitation and reconversion. 100 kHz RF-field strength was used for heteronuclear decoupling using SW$_f$-TPPM during indirect (t$_1$) and direct (t$_2$) detection periods, and continuous wave (CW) during S3 recoupling. A z-filter of 100 μs was inserted before acquisition. The experiments were performed with a recycle delay of 8 s. A total of 352 scans were recorded for a mixing time of 2.5 ms, and 160 scans were recorded for a recoupling time of 7.2 ms.

The double-quantum recoupling efficiency curves for the S3 and BABA recoupling sequences were simulated using the SIMPSON[57] software. The S3 sequence was simulated under varying $^{13}$C-$^{13}$C dipolar distances, while the BABA sequence was simulated under varying $^1$H-$^1$H dipolar distances. The SIMPSON input files for these and related pulse sequences are maintained by Dr. Subhradip Paul and are available on GitHub at https://github.com/dnp-grenoble/simpson, with an archived version available on Zenodo (https://doi.org/10.5281/zenodo.17457240).

## Calculation methods

**Structural optimization and energy calculation.** Our system consisted of the unit cell of UiO-66 (Zr) with the formula [Zr$_6$O$_4$(OH)$_4$(BDC)$_6$]$_4$, where BDC represents 1,4-benzenedicarboxylate. The structure of intact UiO-66 was taken from ref.[58]. We studied both intact UiO-66 and UiO-66 with a dangling linker. We optimized their structures by periodic density functional theory (DFT) simulations using the CP2K package[59]. Perdew-Burke-Ernzerhof (PBE) exchange correlation functional[60] was combined with DZVP-MOLOPT-SR-GTH basis set[61,62]. The Grimme's DFT-D3 semi-empirical method[63] was used to account for the van der Waals interactions. The plane-wave energy cutoff was 400 Ry. The integration over the irreducible Brillouin zone was computed over the Gamma point. Subsequently, the energies were computed by the PBE functional with pcseg-1 basis set[64] for H, C, and O atoms, and TZVP-MOLOPT-SR-GTH basis set for Zr atoms. The Self-Consistent Continuum Solvation (SCCS) implicit solvent model[65,66] was employed. All the CP2K input files are generated by Multiwfn package[67].

**NMR calculation.** We used the Gaussian 16 software to compute $^{13}$C and $^1$H NMR. A part of the energy-optimized structures, the so-called cluster model, was used without periodic boundary conditions to reduce the computational cost. We terminated the surface by adding hydrogen atoms and included a few water molecules inside. The cluster formula is represented by Zr$_6$O$_4$(OH)$_4$(HCOO)$_{22}$·nH$_2$O (n = 1 or 2). The positions of the added hydrogen atoms were optimized while fixing the remaining part, using the b3lyp functional[68] with 6-31 G* for H, C, and O atoms, and LANL2DZ basis set[69] for Zr atoms.

We first computed the isotropic magnetic shielding ($\delta_{calc.}^X$) of $^{13}$C and $^1$H in the optimized structures using the Gauge-Invariant Atomic Orbital (GIAO) scheme[70–73]. The calculations were carried out with the revTPSS functional[74,75], employing pcsSseg-1 basis set[76] for H, C and O atoms, while the LANL2TZ[69,77], basis set was used for Zr atoms. The solvent model density (SMD) continuum solvation model was applied[78]. We then computed the chemical shift ($\delta_{calc.}^X$) through an equation of $\delta_{calc.}^X = \sigma_{calc.}^{ref.} - \sigma_{calc.}^X + \delta_{calc.}^{ref.}$, where "ref." and "exp." represent reference and experiment, respectively. The $\delta_{calc.}^{ref.}$ of 38.5 ppm on $^{13}$C was taken from the methylene of adamantane and the $\delta_{calc.}^{ref.}$ of 1.8 ppm on $^1$H was taken from the adamantane protons.

## Data availability

Data available on request from the authors.

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

## Acknowledgements

G.D.P. acknowledges support by the French National Research Agency through CBH-EUR-GS and the Labex ARCANE (ANR-17-EURE-0003), as well as the European Research Council (ERC-CoG- 2015, No. 682895). K.M. acknowledges support by National Natural Science Foundation of China (22250610195 and 22273083). We acknowledge Prof. Vincent Artero and Dr. Matthieu Koepf for providing access to the chemical laboratory for materials preparation. For the purpose of Open Access, a CC-BY 4.0 public copyright license has been applied by the authors to the present document and will be applied to all subsequent versions up to the Author Accepted Manuscript arising from this submission (https://creativecommons.org/licenses/by/4.0/).

## Author contributions

Conceptualization: Y.F. and G.D.P.. Methodology: Y.F., S.P., Y.Y.. Supervision: K.M., G.D.P.. Writing—review & editing: Y.F., Y.Y., S.P., K.M., G.D.P.

## Competing interests

The authors declare no competing interests.
