## [Transparent Peer Review file · Nature Communications]

Water-generated dangling linkers in a metal-organic framework

Corresponding Author: Professor Gaël De Paëpe

Version 0:

Reviewer comments:

Reviewer #1

(Remarks to the Author)

In this manuscript, De Paëpe et al. elegantly demonstrate the formation of dangling linkers in the presence of high amount of water using solid-state NMR. By employing 1D and 2D DNP NMR experiments and calculations, they successfully localize the water in the system. This work provides new insights into intriguing materials. However, several concerns need to be addressed before the manuscript is ready for publication.

- 1) My main concern is relative to the HETCOR experiment. It was performed with a long mixing time (5 ms). If the goal is to observe close proximity interactions, a shorter mixing time would be more appropriate. Additionally, the C4-OH correlation task is not observed even though the proximity is confirmed by the calculations. Authors note line 94 : « it is interesting and also puzzling to note the absence of cross-peaks between C4 and OH resonances.... ». Could they elaborate on this? Is this due to mobility issues? If so, could a lower temperature experiment help detect this signal?
- 2) The figures 2 for the 2D experiments are too small and should be enlarged for better clarity.
- 3) Regarding the DS-SQ experiment, no correlation between H₂O and OH is observed? Since both are Hydrogen-bonded, shouldn't they correlate in the DQ-SQ? Did the authors try performing the experiment at lower temperature to reduce the mobility? Would increasing the spinning speed or using homonuclear decoupling improve the resolution of this experiment?
- 4) I think that adding a table with experimental and calculated chemical shifts would be beneficial for the reader.
- 5) In the Supporting information, Figure S7, adding a diagonal would likely enhance the clarity for the reader.
- 6) Line 126, « Figure 2c » must be replaced by « Figure 2d »
- 7) Could authors comment on the difference of temperature (mobility) that exists between the experimental and calculation data? Could/should it play a role in the assignment of the experimental data?

Reviewer #2

(Remarks to the Author)

Dear Dr. Davies,

Manuscript NCOMMS-25-51094 reports on an intriguing phenomenon that helps to understand the high stability of specific metal-organic frameworks (MOFs) against water attack. MOFs are an exciting class of porous materials that are discussed for a broad variety of applications. One aspect often preventing commercialisation is the limited stability of many MOFs when exposed to water or humid environments. Therefore, several strategies were developed to prevent water from cleaving the dative bonds between the inorganic building units (IBUs) and the organic linkers, mostly irreversibly. Approaches range from hydrophobisation of the inner surfaces to reduce the water uptake to shielding the IBUs by sterically demanding side chains attached to the linker molecules. A particularly intriguing concept was introduced by Morris et al. in 2018 (Nature Chem. 2018, 10, 1096). By providing sacrificial bonds, the integrity of the MOF topology is protected despite water molecules reacting with the frameworks. This strategy was referred to as the hemilability concept of MOFs, and it was proposed that it might be relevant for several framework types. However, so far, specific examples are rare. Manuscript NCOMMS-25-51094 now presents a study where one carboxylate group of the terephthalate linkers is cleaved but stabilised at the zirconium cluster close to the created open coordination site by multiple hydrogen bonding. The resulting open coordination site then takes up water molecules without destabilising the framework integrity. As both processes occur in the same cluster, the process is reversible solely depending on the water uptake.

Although numerous studies focus on MOF water interactions, the reversibility of the reported observation is remarkable and warrants publication in Nature Communications. Nevertheless, the manuscript should be improved in several places before publication.

The authors seem to regard adsorbed water molecules solely as passive guest molecules. This is pointed out several times, e.g., in the abstract and introduction. It is, however, a view that is too narrow, as, in many cases, water is active, competing with the dative bonds of the linker molecules and the IBUs. In particular, the authors do not seem to be aware of the above-mentioned hemilability concept, which is based on similar mechanisms observed for this study. Thus, I recommend including the concept and broadening the focus of the discussion.

On page one, it is pointed out that "... water could break the linker-metal bonds in UiO-66 under neutral pH and ambient conditions." I wonder what neutral pH means in this context. Water molecules are adsorbed in the samples, and the adsorbed aqueous phase will have different properties compared to bulk water. Strictly speaking, a pH is not defined for the adsorbed water molecules.

On page two, the quantity of adsorbed water molecules is defined as the uptake in the unit cell. Its content is given as $Zr_24O_{24}(BDC)_4$, which is wrong. The unit cell content must be a multiple of UiO-66's stoichiometry, which is $Zr_6O_4(OH)_4(BDC)_4$. To simplify matters, I would use the stoichiometric unit to reference the water uptake.

The ^{13}C MAS NMR spectra for 160 H₂O/u.c. and 320 H₂O/u.c. (Fig. 1c) exhibit similar ratios for resonances 4 and 1, suggesting that the quantity of defect sites is independent of the water uptake in this region. Based on the DNP data, one-sided linker termination could even be quantified to 22%. I find it surprising that the defect site proportions reach a limit already at medium water uptakes, and I would like to know if the authors have an explanation for this observation. It might be worth measuring lower water uptakes as well, where the defect formation is not saturated yet. This would provide access to the underlying thermodynamic properties, offering more profound insight into this phenomenon.

Concerning the 1H-1H DQ NMR spectra. Does the phrase mixing time refer to the DQ excitation time? An excitation time of 57 ms seems relatively short. Such spectra should predominantly sample short distances < 2 Å. Therefore, the three intense resonances for the μ_3 -OH groups are a bit surprising. Their average distance is probably larger than 4 Å. Spimpson simulations, such as those for the ^{13}C - ^{13}C spin pairs, would be helpful, and additional information about the DQ experiment would be appreciated.

To compare the DNP-enhanced NMR spectroscopic experiments to the conventional ones, the water uptake should be similar. I could not find information on how many H₂O, D₂O, and glycerol molecules are adsorbed in the UiO-66 samples before the samples are cooled down for the measurements. Is the uptake of the solvent mixture similar enough to the 160/u.c. sample? Please elaborate on this aspect, and also whether you expect an isotope effect for the one-sided linker displacement. For sure, D₂O will preferentially adsorb to the created open coordination sites. Please also add some information about how far the polarisation can be transferred from the MOF surfaces, with the radicals, into the UiO-66 crystallites, to make sure that chemical variations in surface-near regions do not influence the quantification.

Reviewer #3

(Remarks to the Author)

The results presented by Fu et al. examine the influence of water on the Metal-Ligand interaction within UiO-66, assessing the bonding and de-bonding nature of the carboxylic acid and how this ligand can interact with water molecule(s). They demonstrate new findings using a combination of carefully prepared materials, solid-state NMR, DNP NMR and calculations. Overall the work may be of interest to Nature Communications, however some further details are needed to ensure reproducibility to support their conclusions. Below, a few major and minor comments are provided for the authors to consider.

1. FTIR method requires details on sample preparation.
2. SSNMR methods for MOFs is cited, however, I would recommend the authors add 2 papers from Yining Huang - who has done Zr NMR on UiO and has also demonstrated extensive dynamic effects using ^{13}C and ^{17}O in other systems which are related to the impact SSNMR has had in this field.
3. Can the authors describe the different types of water and u-OH species, it is unclear how different chemical shifts are provided for these functional groups. Perhaps the DFT can provide some guidance to these differences.
4. The authors should not use the terms upfield and downfield in the 1H NMR section; perhaps higher/lower frequencies or shielded/deshielded terminology is more appropriate.
5. American vs British spelling - pick one.
6. ^{13}C - ^{13}C should be superscripted, while Figure 3a, The should be the
7. The T1 values measured for the 1H signals should be provided in the SI (for table for example)...stating these are 5x or more in the text is insufficient as the authors make this claim a few times.
8. References to the NMR section should be added such as the references used for 1H/ ^{13}C , BABA sequence.
9. Does the glycerol/D₂O/H₂O influence the MOF structure/water interactions the authors report...the solution contains 40% water - does this impact the ligand - water / ligand-metal interaction?
10. Full experimental conditions are needed for the ^{13}C - ^{13}C DQ-SQ DNP MAS NMR results, including recycle delay, transients, pulse powers (which pulse sequence), etc.
11. The SIMPSON input file should be provided in the SI with the pulse program implemented.

12. Figure S10 states DNP-enhanced but the experimental/test indicates this was a bloch pulse experiment using SSNMR with a recycle delay of 250s. I suspect this is incorrect, as direct ^{13}C DNP NMR would likely not be effective using Amupol under these sample conditions, while the nuclear T1 for ^{13}C would be much longer than 250s at 100 K.

Version 1:

Reviewer comments:

Reviewer #1

(Remarks to the Author)

The manuscript is now ready for publication

Reviewer #2

(Remarks to the Author)

The authors have thoroughly revised the manuscript NCOMMS-25-51094, addressing the raised questions and concerns satisfactorily. I suggest publishing the revised manuscript without further corrections.

RESPONSE TO REVIEWERS' COMMENTS

Reviewer #1 (Remarks to the Author):

In this manuscript, De Paëpe et al. elegantly demonstrate the formation of dangling linkers in the presence of high amount of water using solid-state NMR. By employing 1D and 2D DNP NMR experiments and calculations, they successfully localize the water in the system. This work provides new insights into intriguing materials. However, several concerns need to be addressed before the manuscript is ready for publication.

1) My main concern is relative to the HETCOR experiment. It was performed with a long mixing time (5 ms). If the goal is to observe close proximity interactions, a shorter mixing time would be more appropriate. Additionally, the C4-OH correlation peak is not observed even though the proximity is confirmed by the calculations. Authors note line 94 : « it is interesting and also puzzling to note the absence of cross-peaks between C4 and OH resonances... ». Could they elaborate on this? Is this due to mobility issues? If so, could a lower temperature experiment help detect this signal?

Answer: Thank you for your valuable comment. We have also acquired a HETCOR spectrum with a shorter mixing time of 1 ms (see Figure S4), which shows similar results to those obtained at 5 ms. For the main manuscript, we have retained the 5 ms spectrum due to its higher sensitivity, while the 1 ms spectrum has been added to the Supporting Information for comparison.

Also, we realize that the wording we use was not ideal. There is nothing “puzzling” in our results since we are able to fully explain the absence of cross peak between C4 and OH resonances between 1 to 3 ppm (see point 3 below for more details). Thus, we have corrected the text and improve the wording to avoid confusion.

2) The figures 2 for the 2D experiments are too small and should be enlarged for better clarity.

Answer: Thank you for your comment. 2D experiments including 1H-1H 2D experiment and 13C-1H HETCOR experiment in Figure 2 have been enlarged to improve clarity, as suggested.

3) Regarding the DS-SQ experiment, no correlation between H₂O and OH is observed? Since both are Hydrogen-bonded, shouldn't they correlate in the DQ-SQ? Did the authors try performing the experiment at lower temperature to reduce the mobility? Would increasing the spinning speed or using homonuclear decoupling improve the resolution of this experiment?

Answer: Thank you for your comment. In fact, the H₂O molecules engaged in hydrogen bonding do show correlations with the μ_3 -OH groups in Figure 2c. In the figure, the green signal (around 6 ppm) corresponds to H₂O involved in hydrogen bonding, while the orange signal represents neighboring μ_3 -OH groups, which also shift to approximately 6 ppm due to hydrogen bonding with dangling linkers, as illustrated in the A-I structure of Figure 3a. This interaction contributes to stabilizing the formation of the dangling linker. Because these signals have very similar chemical shifts, their correlations overlap in the spectrum. To improve clarity, we have enlarged Figure 2c so that these correlations are more distinctly visible.

4) I think that adding a table with experimental and calculated chemical shifts would be beneficial for the reader.

Answer: Thank you for your suggestion. We have added two tables, S2 and S3, in the Supporting Information, which compare the experimental and calculated chemical shifts for carbons and protons.

5) In the Supporting information, Figure S7, adding a diagonal would likely enhance the clarity fir the reader.

Answer: Thank you for your comment. A diagonal line has been added to previous Figure S7 (now Figure S10) to improve clarity.

6) Line 126, « Figure 2c » must be replace by « Figure 2d »

Answer: Thank you for pointing this out. We have corrected this typo.

7) Could authors comment on the difference of temperature (mobility) that exists between the experimental and calculations data? Could/should it play a role in the assignment of the experimental data?

Answer: DNP-enhanced spectra were obtained at low temperature (~100 K) and show a consistent chemical shift for C4 compared with the experimental spectra recorded at 298 K. Moreover, both the DNP-enhanced spectra at 100 K and the DFT calculations at 0 K yield consistent relative chemical shifts between C4 and C1. Therefore, we believe that temperature and molecular mobility do not substantially influence our conclusions.

Reviewer #2 (Remarks to the Author):

Dear Dr. Davies,

Manuscript NCOMMS-25-51094 reports on an intriguing phenomenon that helps to understand the high stability of specific metal-organic frameworks (MOFs) against water attack. MOFs are an exciting class of porous materials that are discussed for a broad variety of applications. One aspect often preventing commercialisation is the limited stability of many MOFs when exposed to water or humid environments. Therefore, several strategies were developed to prevent water from cleaving the dative bonds between the inorganic building units (IBUs) and the organic linkers, mostly irreversibly. Approaches range from hydrophobisation of the inner surfaces to reduce the water uptake to shielding the IBUs by sterically demanding side chains attached to the linker molecules. A particularly intriguing concept was introduced by Morris et al. in 2018 (Nature Chem. 2018, 10, 1096). By providing sacrificial bonds, the integrity of the MOF topology is protected despite water molecules reacting with the frameworks. This strategy was referred to as the hemilability concept of MOFs, and it was proposed that it might be relevant for several framework types. However, so far, specific examples are rare. Manuscript NCOMMS-25-51094 now presents a study where one carboxylate group of the terephthalate linkers is cleaved but stabilised at the zirconium cluster close to the created open coordination site by multiple hydrogen bonding. The resulting open coordination site then takes up water molecules without destabilising the framework integrity. As both processes occur in the same cluster, the process is reversible solely depending on the water uptake.

Although numerous studies focus on MOF water interactions, the reversibility of the reported observation is remarkable and warrants publication in Nature Communications. Nevertheless, the manuscript should be improved in several places before publication.

The authors seem to regard adsorbed water molecules solely as passive guest molecules. This is pointed out several times, e.g., in the abstract and introduction. It is, however, a view that is too narrow, as, in many cases, water is active, competing with the dative bonds of the linker molecules and the IBUs. In particular, the authors do not seem to be aware of the above-mentioned hemilability concept, which is based on similar mechanisms observed for this study. Thus, I recommend including the concept and broadening the focus of the discussion.

Answer: Thank you for your insightful comment. We have revised the abstract, introduction and discussion part to incorporate the concept of hemilability and broaden the discussion on the role of water, in line with your suggestion.

On page one, it is pointed out that "... water could break the linker-metal bonds in UiO-66 under neutral pH and ambient conditions." I wonder what neutral pH means in this context. Water molecules are adsorbed in the samples, and the adsorbed aqueous phase will have different properties compared to bulk water. Strictly speaking, a pH is not defined for the adsorbed water molecules.

Answer: Thank you for pointing this out. The sentence has been revised to: "Water molecules could break the linker-metal bonds in UiO-66 under ambient conditions."

On page two, the quantity of adsorbed water molecules is defined as the uptake in the unit cell. Its content is given as $Zr_{24}O_{24}(BDC)_{24}$, which is wrong. The unit cell content must be a multiple of UiO-66's stoichiometry, which is $Zr_6O_4(OH)_4(BDC)_24$. To simplify matters, I would use the stoichiometric unit to reference the water uptake.

Answer: Thank you for pointing this out. We initially used the unit cell formula $Zr_{24}O_{24}(BDC)_{24}$ to represent the activated UiO-66 structure. According to the literature (W. Zhou *et al.*, *J. Am. Chem. Soc.*, 2013, 135, 10525–10532), upon full activation at high temperature (~250 °C) under vacuum, each $Zr_6O_4(OH)_4$ cluster loses two H₂O molecules, resulting in a Zr_6O_6 node. However, since a significant number of μ -OH groups remain in the structure, we chose to describe the unit cell as $Zr_{24}O_{16}(OH)_{16}(BDC)_{24}$ in the text. We appreciate your careful observation and clarification.

The ¹³C MAS NMR spectra for 160 H₂O/u.c. and 320 H₂O/u.c. (Fig. 1c) exhibit similar ratios for resonances 4 and 1, suggesting that the quantity of defect sites is independent of the water uptake in this region. Based on the DNP data, one-sided linker termination could even be quantified to 22%. I find it surprising that the defect site proportions reach a limit already at medium water uptakes, and I would like to know if the authors have an explanation for this

observation. It might be worth measuring lower water uptakes as well, where the defect formation is not saturated yet. This would provide access to the underlying thermodynamic properties, offering more profound insight into this phenomenon.

Answer: Thank you for your comment. We attribute the plateau in defect site proportion at medium water uptakes to the fact that the initial water molecules predominantly interact with the MOF pore walls. As a result, the intensity of the dangling linker-related carboxylate groups (C4) in the 160 H₂O/u.c. and 320 H₂O/u.c. samples, relative to the original carboxylate signals (C1) in the CP experiments, remains nearly constant. While CP is not strictly quantitative, the same contact time was used for both samples, and since both the C4 and C1 signals arise from carboxylates, the peak intensity ratio provides a reliable basis for comparison.

After the MOF pores are saturated with water, additional molecules are likely confined within the pores and do not interact directly with the ligands or pore walls. We agree that studying low-defect structures would be valuable; however, the low defect concentration makes it difficult to obtain a clear signal, even with DNP.

Concerning the 1H-1H DQ NMR spectra. Does the phrase mixing time refer to the DQ excitation time? An excitation time of 57 μ s seems relatively short. Such spectra should predominantly sample short distances $< 2 \text{ \AA}$. Therefore, the three intense resonances for the μ_3 -OH groups are a bit surprising. Their average distance is probably larger than 4 \AA . Simpson simulations, such as those for the 13C-13C spin pairs, would be helpful, and additional information about the DQ experiment would be appreciated.

Answer: We thank the reviewer for raising this point regarding the short mixing time. The 57 μ s reported in the manuscript corresponds to the total mixing (recoupling) time, which includes both the excitation and reconversion periods of the DQ experiment. We intentionally chose a short mixing time to minimize relayed transfers and ensure that the observed correlations primarily reflect direct, short-range 1H-1H interactions.

As shown in our accompanying SIMPSON simulations (Figure S7, shown below), the transfer between aromatic protons separated by $\sim 2.5 \text{ \AA}$ is predicted to be about two orders of magnitude stronger than that between μ_3 -OH protons ($\sim 5.6 \text{ \AA}$ apart). This trend is consistent with the experimental DQ-SQ spectrum in Figure 2c, where the autocorrelation peak of the aromatic protons is significantly more intense than that of the μ_3 -OH group.

For completeness, we also performed measurements with a longer recoupling time of 171 μ s (Figure S6, shown below). The resulting spectra were essentially identical to those obtained with the 57 μ s mixing time, confirming that longer mixing does not introduce additional correlations under our conditions. However, the 171 μ s experiment exhibited lower overall sensitivity despite the same number of scans (160), so we chose to present the 57 μ s data in the main text and include the 171 μ s spectrum in the Supporting Information for comparison.

Recoupling Time = 171 μs

To compare the DNP-enhanced NMR spectroscopic experiments to the conventional ones, the water uptake should be similar. I could not find information on how many H₂O, D₂O, and glycerol molecules are adsorbed in the UiO-66 samples before the samples are cooled down for the measurements. Is the uptake of the solvent mixture similar enough to the 160/u.c. sample? Please elaborate on this aspect, and also whether you expect an isotope effect for the one-sided linker displacement. For sure, D₂O will preferentially adsorb to the created open coordination sites. Please also add some information about how far the polarisation can be transferred from the MOF surfaces, with the radicals, into the UiO-66 crystallites, to make sure that chemical variations in surface-near regions do not influence the quantification.

Answer: Thank you for your comments. For the DNP sample preparation, approximately 45 μL of the DNP juice (Glycerol/ D_2O / H_2O = 6:3:1) was added to the dry UiO-66 sample (around 30 mg). This corresponds to an estimated water content of about 55 H_2O molecules and 165 D_2O per unit cell of $\text{Zr}_{24}\text{O}_{16}(\text{OH})_{16}(\text{BDC})_{24}$. This value lies between the medium and high-water uptake regimes (approximately 160 and 320 H_2O per unit cell, respectively), and is therefore comparable to the 160 H_2O /u.c. reference sample.

Given the similar hydrogen-bonding behavior of D_2O and H_2O , we do not expect a significant isotope effect on the one-sided linker displacement. Although D_2O may preferentially adsorb at open coordination sites, its overall interaction with the framework should be comparable to that of H_2O .

Hyperpolarization can be transferred from the biradicals into the UiO-66 crystallites via ^1H - ^1H spin diffusion. Depending on the ^1H T_1 relaxation time inside the MOF, this ^1H hyperpolarization can be transported over hundreds of nanometer length scales. Therefore, polarization can reach well into the interior of the MOF particles, which reduces the influence of surface-localized chemical variations on the NMR quantification. Moreover, all quantitative data discussed in the main text were obtained from conventional NMR measurements, ensuring reliable quantification independent of DNP effects.

Reviewer #3 (Remarks to the Author):

The results presented by Fu et al. examine the influence of water on the Metal-Ligand interaction within UiO-66, assessing the bonding and de-bonding nature of the carboxylic acid and how this ligand can interact with water molecule(s). They demonstrate new findings using a combination of carefully prepared materials, solid-state NMR, DNP NMR and calculations. Overall the work may be of interest to Nature Communications, however some further details are needed to ensure reproducibility to support their conclusions. Below, a few major and minor comments are provided for the authors to consider.

1. FTIR method requires details on sample preparation.

Answer: Thank you for your helpful suggestion. We have added details on the sample preparation procedure for the FT-IR method in SI.

2. SSNMR methods for MOFs is cited, however, I would recommend the authors add 2 papers from Yining Huang - who has done Zr NMR on UiO and has also demonstrated extensive dynamic effects using ^{13}C and ^{17}O in other systems which are related to the impact SSNMR has had in this field.

Answer: We have now cited the relevant papers from Prof. Huang as recommended.

3. Can the authors describe the different types of water and μ -OH species, it is unclear how different chemical shifts are provided for these functional groups. Perhaps the DFT can provide some guidance to these differences.

Answer: The DFT-calculated ^1H chemical shifts for all μ_3 -OH groups across the different configurations are provided in Table S3 of the Supporting Information. Additionally, Table S4 presents the DFT-calculated ^1H chemical shifts for water-related protons. Together, these data allow clear differentiation between the various μ -OH groups and water species in the system.

4. The authors should not use the terms upfield and downfield in the ^1H NMR section; perhaps higher/lower frequencies or shielded/deshielded terminology is more appropriate.

Answer: We have revised the text accordingly, replacing the terms upfield and downfield with higher/lower frequencies as appropriate.

5. American vs British spelling - pick one.

Answer: We have revised the manuscript to use American spelling throughout.

6. ^{13}C - ^{13}C should be superscripted, while Figure 3a, The should be the

Answer: Thank you for pointing out. We have corrected the mistakes.

7. The T_1 values measured for the ^1H signals should be provided in the SI (for table for example)...stating these are 5x or more in the text is insufficient as the authors make this claim a few times.

Answer: We have now included the measured ^1H T_1 values for all ^1H signals across all samples in Table S1 of the Supporting Information.

8. References to the NMR section should be added such as the references used for $^1\text{H}/^{13}\text{C}$, BABA sequence.

Answer: We have now included references to the NMR section in the text.

9. Does the glycerol/ $\text{D}_2\text{O}/\text{H}_2\text{O}$ influence the MOF structure/water interactions the authors report...the solution contains 40% water - does this impact the ligand - water / ligand-metal interaction?

Answer: Thank you for raising this important point. Although the DNP juice contains only 40% water, the observed ^{13}C chemical shift for the C4 peak remains unchanged compared to

measurements in pure water. This consistency suggests that the glycerol/D₂O/H₂O mixture does not significantly affect the ligand-water or ligand-metal interactions within the MOF structure.

10. Full experimental conditions are needed for the ¹³C-¹³C DQ-SQ DNP MAS NMR results, including recycle delay, transients, pulse powers (which pulse sequence), etc.

Answer: We have now included the full experimental conditions for the ¹³C-¹³C DQ-SQ DNP MAS NMR measurements in the SI SSNMR method section.

11. The SIMPSON input file should be provided in the SI with the pulse program implemented.

Answer: Thank you for your suggestion. We have included a link to the SIMPSON input file in the Supporting Information (SI). The SIMPSON input files for various pulse sequences are maintained by Dr. Subhradip Paul and are available at the following GitHub repository: <https://github.com/dnp-grenoble/simpson>

12. Figure S10 states DNP-enhanced but the experimental/test indicates this was a bloch pulse experiment using SSNMR with a recycle delay of 250s. I suspect this is incorrect, as direct ¹³C DNP NMR would likely not be effective using Amupol under these sample conditions, while the nuclear T₁ for ¹³C would be much longer than 250s at 100 K.

Answer: Thank you for raising this important point. To ensure the quantitative accuracy of our experiment, we synthesized a MOF sample with ¹³C-labeled carboxylate groups of linker. For the obtaining wet UiO-66, the direct polarization experiments, we set the recycle delay to 200 s, given a measured ¹³C T₁ of 38 s at room temperature. The resulting spectrum accurately

reflects the expected intensity ratio of the two carboxylate groups. We have updated the results in the text and SI Figure S13.